# A Novel Fruit Shape Classification Method: BLS-Levelset

Dan Zhang
*College of Mechanical and Electrical Engineering*
*Dalian Minzu University*
Dalian,China
zhangdan@dlnu.edu.cn

Tieshan Li
*College of Automation Engineering*
*University of Electronic Science and Technology*
Chengdu,China
tieshanli@126.com

C.L. Philip Chen
*College of Computer Science and Engineering*
*South China University of Technology*
Guangzhou,China
philio.chen@ieee.org

Yi Zuo
*College of Navigation*
*Dalian Maritime University*
Dalian,China
zuo@dlmu.edu.cn

*Abstract*—**Although some fruits have a fixed shape, but the growth process, the shape will still be different, different fruit shape has a certain impact on the subsequent fruit classification and selection. In order to realize fruit shape classification quickly and accurately, a novel fruit shape classification method based on level set and broad learning system(BLS) is proposed in this paper. Firstly, fruit shape information is extracted based on level set, then fruit image information is extracted based on BLS, and finally, shape information and fruit information are fused to carry out fruit shape classification research. The method was applied to the classification of the fruit shape, and the experimental results showed the effectiveness of the proposed method.**

*Keywords—fruit shape classification, broad learning system; level set*

## I. INTRODUCTION

In fruit shape classification and segmentation, image processing, computer vision, artificial intelligence and other technologies have been widely used. The development of deep learning has further promoted the application of neural networks in agriculture. Image classification and recognition methods have been widely used in agriculture, and the early methods mainly focus on random forest [1], support vector machine [2], clustering [3] and so on.Wei et al. [4] proposed an OSTU method based on fruit color recognition, the recognition accuracy is about 95%, but the color based recognition is very sensitive to changes in the external scene, and the algorithm is not robust. Recognition methods based on shape [5] and texture [6] are also emerging. Meng et al. [7] used the edge detection method to extract the target object, and used the Canny operator and Hoff transform to detect the target contour to achieve positioning. Chaivivatrakul et al. [8] proposed the use of texture features for fruit detection, and the recognition accuracy was above 90%. Due to the limited ability of single features to express fruit characteristics in complex environments, methods based on feature fusion have gradually emerged. The method of feature fusion is to combine different features to distinguish different targets. Zhuang et al. [9] combined information such as color, texture and shape to identify fruits and guide fruit picking robots. Payne et al. [10] used color segmentation and texture of RGB and YCbCr to segment fruit targets, but when the color features were not obvious, the recognition accuracy was low. Kim et al. [11] proposed a fruit recognition method with shared feature set, and the fruit accuracy rate can reach 90%. The level set method is also widely used in fruit shape classification. Gui et al. [12] proposed an apple shape classification method based on level set and motion estimation. Qiu[13] et al proposed an improved multi-level set C-V model for the classification of red pine seeds. Although the level set method has a good effect on fruit shape segmentation, it is still difficult to process large data sets. In recent years, the method based on the combination of level set and neural network has been widely used in agriculture.

Deep learning has been widely used in various fields and achieved great success. Hussain et al. [14] proposed a framework for automatic detection and recognition of fruits and vegetables based on deep learning. Alresheedi et al. [15] used classical feature fusion into deep learning to identify jujube fruits. Ji et al. [16] adopted a lightweight deep network to achieve better identification of Apple. Sun et al. [17] combined DeepLab ResNet semantic segmentation and level set shape constraint to detect fruit tree flowers. Due to the strong feature extraction capability of deep learning, it has also been widely used in agriculture and achieved good results in segmentation accuracy, such as [18-20]. However, in the application process of deep learning, there are also problems such as long network training time, multiple training parameters, large data demand, parameter adjustment in network transplantation, and challenges in actual application scenarios. Researchers also need to pay attention to the development of technology and algorithms, and constantly improve the effectiveness and adaptability of algorithms.

Broad learning system [21-22] was proposed in 2017-2018. This network structure requires fewer calculation parameters, has high efficiency and has good image classification effect. BLS has been well applied in many aspects of image

classification and detection, such as [23-28]. It can be seen that BLS has a wide range of applications and broad application prospects.

Xanthoceras sorbifolium is the only biomass energy tree species suitable for development in northern China determined by the National Forestry and Grassland Administration, and it is a key woody oil tree species in China. Rich in unsaturated fatty acids, seed kernel oil is a new type of healthy woody edible vegetable oil and an important tree species for precise poverty alleviation, rural revitalization and ecological construction in northern China. Fruit shape is different in the growth process, and there is a phenomenon of seed scattering caused by cracking. Fruit shape also has a certain influence on the subsequent processing and extraction of seed kernel. This paper is based on the natural acquisition of the fruit image data set, which is used to test the method in this paper.

In this paper, a novel fruit shape classification method combining the advantages of level set method and BLS is proposed to solve the problems such as the difficulty of level set method to handle data set effectively, the long training time and the large computation amount of deep learning method.

## II. RELATED WORK

### A. Level Set

In 1988, the set deformation level set method was introduced by Osher and Sethian. This approach primarily represents a low-dimensional evolving curve or surface using a high-dimensional functional surface. Essentially, the evolving curve or surface (interface) is conveyed through the zero level set of the high-dimensional level set function. As a result, the evolution equation governing the curve or surface is reformulated into a partial differential equation pertaining to the high-dimensional level set function, while the moving boundary surface is identified by solving the equation related to the level set function. Following this, various scholars have refined the level set method. One such improvement is the region scalable fitting energy (RSF) model proposed by Li [29], which employs statistical analysis of local brightness information to better approximate the target and enhance image segmentation accuracy. In this study, this method was utilized to extract information regarding the shape of fruits.

The RSF model divides the image into two parts. One part is the outside of the given initial contour $C$, represented by $\Omega_1$, and the other part is the inside of the contour $C$, represented by $\Omega_2$, represented by a function representing the image intensity of the two regions. Gaussian kernel function is used to control local properties.

The functions representing the inner and outer regions of the contour are $f_1(x)$ and $f_2(x)$. The Gaussian kernel function is:

$$K_\sigma(u) = \frac{1}{2\pi\sigma^2} c^{-\frac{|u|}{2\sigma^2}} \quad (1)$$

The energy curve is expressed as:

Identify applicable funding agency here. If none, delete this text box.

$$E(C, f_1(x), f_2(x)) = \sum_{i=1}^{2} \lambda_i \int_{\Omega_i} K(x-y)|I(y) - I(x)|^2 \, dy \quad (2)$$

Where $\lambda$ is a constant greater than zero and $I$ represents the image. Let the length of contour $C$ be $|C|$, in order to get a smoother contour, the energy function is expressed as:

$$(C, f_1(x), f_2(x)) = \int E(C, f_1(x), f_2(x))dx + v|C| \quad (3)$$

The above function is topologically transformed, and the expression of the level set is as follows:

$$E(C, f_1(x), f_2(x)) = \sum_{i=1}^{2} \lambda_i \int K(x-y)|I(y) - f_i(x)|^2 M_i(\phi(y))dy + \frac{1}{2}\mu \int (|\nabla\phi(x)| - 1)^2 dx + v \int |\nabla H_\varepsilon(\phi(x))| dx \quad (4)$$

Where $M_1(\phi(y)) = H(\phi(y)), M_2(\phi(y)) = 1 - H(\phi(y))$.

Use the smoothing function definition $H_\varepsilon$:

$$H_\varepsilon(x) = \frac{1}{2}\left[1 + \frac{2}{\pi}\arctan(\frac{x}{\varepsilon})\right] \quad (5)$$

Its derivative is:

$$\delta_\varepsilon(x) = H_\varepsilon'(x) = \frac{1}{\pi}\frac{\varepsilon}{\varepsilon^2 + x^2} \quad (6)$$

Using gradient descent method to calculate the minimum energy function, we get:

$$\frac{\partial\phi}{\partial t} = -\delta_\varepsilon(\phi)(\lambda_1 e_1 - \lambda_2 e_2) + v\delta_\varepsilon(\phi)div(\frac{\nabla\phi}{|\nabla\phi|}) + \mu(\nabla^2\phi - div(\frac{\nabla\phi}{|\nabla\phi|})) \quad (7)$$

Where $v$ and $\mu$ are constants. $e_i = \int K_\sigma(y-x)|I(x) - f_i(y)|^2 dy, i = 1, 2$. Equation (7) is the level set evolution equation of RSF algorithm. This method can obtain the fruit outline of the corymbophyllum.

### B. Broad Learning System

The structure of the BLS is simple and flexible, and it can carry out incremental learning without repeated calculation. Set the input data to $X$, $X \in \mathbb{R}^{m \times n}$. The input layer of BLS consists of feature nodes and enhancement nodes. Suppose $Z$ is used to represent the feature node, $H$ is used to represent the enhanced node, and their activation functions are respectively represented by $\phi$ and $\xi$, then the $ith$ feature node transformation and $jth$ enhanced node transformation are applied, $Z_i = \phi_i(XW_{ei} + \beta_{ei})$, $H_j = \xi_j(Z^n W_{hj} + \beta_{hj})$,

$Z^n = [Z_1, Z_2, ..., Z_n]$ . Where $W_{ei}$ and $W_{hj}$ are random weights. $\beta_{ei}$ and $\beta_{hj}$ are random biases. For groups of enhanced nodes there are $H^m = [H_1, H_2, ..., H_m]$ . Let the input data at this time be represented by $A$ , then $A = [Z^n | H^m]$ . Suppose the output data is $Y$ , in this case, the network weights are $W$ , then, $Y = AW$ . BLS is solved by pseudo-inverse, and $W$ can be approximated by $A^+$ , $W = [Z^n | H^m]^+ Y$ .

The BLS can carry out incremental computation, which can add feature nodes, enhance nodes, or both at the same time. Suppose that the number of enhanced nodes is $q$ , the original node feature is represented by $A^m$ , and the node after adding nodes is $A^{m+1}$ , then,

$$A^{m+1} = [A^m | \xi(Z^n W_{h_{m+1}} + \beta_{h_{m+1}})] \tag{8}$$

Where $W_{h_{m+1}}$ and $\beta_{h_{m+1}}$ are randomly generated, then calculate the weights, they are necessary to use the following formula:

$$A^{m+1} = \begin{bmatrix} (A^m)^+ - DB \\ B^T \end{bmatrix} \tag{9}$$

Where $D = (A^m)^+ + \xi(Z^n W_{h_{m+1}} + \beta_{h_{m+1}})$ ,

$$B^T = \begin{cases} (C)^+ & \text{if C} \neq 0 \\ (1 + D^T D)^{-1} B^T (A^m)^+ & \text{if C} = 0 \end{cases} \tag{10}$$

$C = \xi(Z^n W_{h_{m+1}} + \beta_{h_{m+1}}) - A^m D$ . The weights are:

$$W^{m+1} = \begin{bmatrix} W^m - DB^T Y \\ B^T Y \end{bmatrix} \tag{11}$$

From the above deduction, it can be seen that in the process of incremental learning, the original data is not repeated, and this network structure and calculation method greatly improve the computing efficiency. Therefore, BLS is efficient in data processing.

### III. PROPOSED METHOD

Fruit images are generally collected in natural environment, and there are influences such as occlusion and illumination, so it is very difficult to achieve accurate fruit shape classification and ensure real-time performance. This paper innovatively combines level set fruit contour extraction with BLS for fruit classification, which can not only accurately extract fruit contour information, but also make full use of the quickness of BLS. A fruit classification method is proposed: BLS-Levelset. Firstly, the fruit contour information was obtained based on the level set method (RSF) and processed into node features.

Secondly, the node feature with contour is fused with the node feature, and the node feature with contour is obtained by using random function transformation. Finally, the enhanced node features are generated by random contour node features, and all the enhanced nodes and feature nodes are sent to the output layer together.

In this paper, the self-made Xanthoceras sorbifolium data set was used as the fruit shape classification data set. In the process of growth, the Xanthoceras sorbifolium fruit mostly forms round, oval, and other shapes. We divided the shapes of Xanthoceras sorbifolium fruit into three categories, as shown in Figure 1(a).

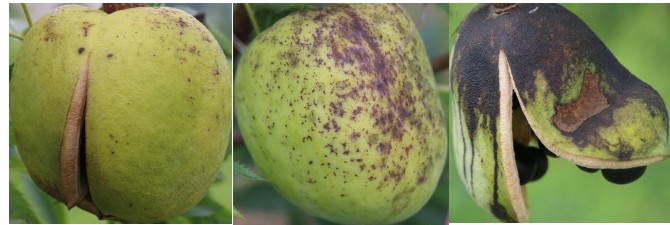

(a) Fruit image: round, oval, other shapes

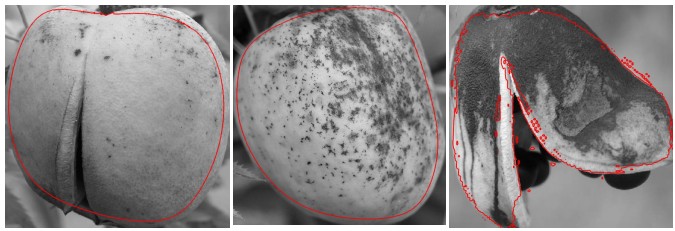

(b) Image of fruit shape segmentation contour

Fig. 1 Fruit and corresponding segmentation images

Suppose the input data is $X$ , the output data is $Y$ , they all belong to $\mathbb{R}^D$ .Firstly, the input data, namely, fruit shape, was segmented by level set RSF method to obtain fruit contour data. The segmentation results of three types of fruit shape data were shown in Figure 1(b). For input data $x_i$ , $i = 1, 2..., n$ . The approximate contour data of fruit shape obtained by level set segmentation method are as follows:

$$f_{RSF}(x_i) = I(\frac{\partial \phi(x_1)}{\partial t}), ..., I(\frac{\partial \phi(x_i)}{\partial t}) \tag{12}$$

Where $I$ represents an image with a segmented contour, and for data $X$ , the resulting contour features are:

$$F = \begin{pmatrix} f(x_1) \\ ... \\ f(x_n) \end{pmatrix} \tag{13}$$

Then the obtained contour information is preprocessed and the normalized data $F_{RSF}$ is obtained, The contour information is fused with the image data and sent to the feature node to obtain the randomly activated feature node data $F_{RSFI}$ .Then, the

$F_{RSFI}$ are randomly mapped to generate group feature nodes. Set the mapping function as $\varphi$, then $ith$ group feature nodes are:

$$Z_{iRSF} = \varphi_i = (F_{RSFI}W_{ei} + \beta_{ei}), i = 1,...n \quad (14)$$

Where $W_{ei}$ is the weight and $\beta_{ei}$ is the bias, they are all randomly generated, and all feature nodes can be represented as $Z_{RSFI}^n = [Z_{RSFI1},...Z_{RSFIn}]$.

For enhanced nodes, $m$ groups, then, $H_j = \xi_j(Z_{RSFI}^n W_{hj} + \beta_{hj}), j = 1,...,m$, where $W_{hj}$ and $\beta_{hj}$ are all randomly generated, all enhanced nodes are represented as $H_{RSF}^m = [H_1,...H_m]$. Finally, the feature node and the enhancement node are input to the output layer, and the following results are obtained:

$$Y = [Z_{RFSI}^n \mid H_{RFS}^m]W^m \quad (15)$$

Where $W^m$ is the solution to the following optimization problem:

$$\arg\min_{W_m} : \left\| [Z_{RFS}^n \mid H_{RFS}^m]W^m - Y \right\|_2^2 + \lambda \left\| W_m \right\|_2^2 \quad (16)$$

$W^m$ are solved by pseudo-inverse matrix.

## IV. Experiment and Analysis

We selected the self-made fruit shape classification data set information experiment of Wenguan fruit. All images were naturally collected from Wenguan Fruit practice base. A total of 1069 images were collected, of which 657 were set as training set and 412 were set as test set. The fruit shapes were divided into three categories: round, oval and other shapes. In order to demonstrate the effectiveness of the proposed method, several methods were used to compare it with BLS-Levelset, including K-NN[30], DBN[31], and BLS, Fuzzy_BLS[32].

### A. Experimental Settings

The detailed classification of the fruit shapes is shown in Fig. 1. The specific parameter Settings are as follows. In order to obtain a better classification effect, we use the search method to find the combination of nodes. The search range is set as the number of feature nodes [1,20], the search range of the number of feature node Windows [1,20], and the step size is 1; the search range of enhanced nodes is [2,200], and the step size is 10. The sparse representation regularization parameter in width learning is set to 0.8, and the contraction parameter of the enhancement node is set to 0.8. The level set method iteration test is set to 40. Other comparison methods used the relevant parameters of the original method.

### B. Comparison of Classification results

In this paper, a search algorithm is used to find the optimal node for BLS. $N_1$ is the number of feature nodes per window, $N_2$ is the number of feature node Windows, and $N_3$ is the number of enhanced nodes. This paper mainly uses classification accuracy as the evaluation criterion, that is, the percentage of all correctly classified quantities in the total quantity. In addition, this paper also compares the running time of the algorithm (unit: s). The specific classification of each method is shown in Table I.

TABLE I. The performance of each method

| Merhods | Test_Acc |
|---|---|
| BLS | 62.14% |
| DBN | 57.77% |
| K-NN | 58.50% |
| Fuzzy_BLS | 58.50% |
| BLS-Levelset | 62.38% |

It can be seen from the table that the accuracy of the method based on broad learning is higher than other methods. The nodes of BLS search are 1, 19, 32. The search nodes of the BLS-Levelset method are 1, 1, 22. It can be seen from the table that the proposed method obtains the highest classification accuracy, which shows the effectiveness of the proposed method. However, it can be seen from the table that the classification accuracy of many methods is not high, mainly because of the complexity of the naturally collected fruit images, which have the influence of occlusion and light, and many cracks of the fruit also have great interference with the shape classification. In the future, it is hoped that the application of natural acquisition image will further strengthen the research of anti-interference feature extraction method.

## V. Conclusion

This paper proposes a fruit shape classification method: BLS-Levelset. The proposed method is fully integrated with the fruit shape classification task, and the fruit contour information extracted based on the level set method is effectively fused with the image information. In this method, the fruit shape information is effectively represented in the node features and the expression of features is optimized. The experimental results show that the method proposed in this paper can achieve good results on the fruit shape classification data set through the fusion of contour features. In the future, we hope that more feature optimization methods can be used in width learning and fruit shape classification to overcome the influence of occlusion and lighting, and further improve the effect of natural acquisition image fruit shape classification.

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
