# OpenReview forum: "A Novel Fruit Shape Classification Method: BLS-Levelset"
_IEEE.org/ICIST/2024/Conference — IEEE ICIST 2024 Conference Submission_

### Official Review · Reviewer_FAKY · 2024-08-21
**This article is very interesting and a good one**

**Rating:** 7
**Confidence:** 3

**Review:**

In this paper, a novel fruit shape classification method based on level set and BLS was proposed to realize fruit shape classification quickly and accurately. The obtained result is valuable and can be accepted if the following problems can be clarified.
(1)	In the INTRODUCTION, the shortages of those relevant studies are suggested to be further summarized.
(2)	In the end of Section 1, the organization of this study is suggested to be summarized.
(3)	There exist several spelling and grammar errors. Please check carefully and further polish.
(4)	In the EXPERIMENT AND ANALYSIS, more analysis can be added to better explain the main results of this paper, that's not enough.

---

### Official Review · Reviewer_WYXj · 2024-08-22
**The work contributes to the field by proposing an efficient and relatively accurate method for fruit shape classification, which can benefit tasks like automated fruit harvesting and quality inspection.**

**Rating:** 9
**Confidence:** 4

**Review:**

this article presents an interesting and novel fruit shape classification method that effectively combines contour-based and image-based features. The integration of level set and BLS shows promising results, highlighting the potential of such hybrid approaches in agricultural image analysis. However, further improvements are needed to enhance the method's robustness in complex natural environments. The work contributes to the field by proposing an efficient and relatively accurate method for fruit shape classification, which can benefit tasks like automated fruit harvesting and quality inspection. Below is a list of comments that should be taken into account further when revising the paper.
1.	There are a few typos in this paper which should be corrected. And there are some notions missed. Please make some corrections.
2.	Please add the necessary comments for Figures.
3.	Look out for the following grammatical and spelling errors: misspelling，redundant or unnecessary words，subject-predicate consistency problem，punctuation error，preposition error.

---

### Official Review · Reviewer_zCP6 · 2024-08-26
**A Novel Fruit Shape Classification Method: BLS-Levelset**

**Rating:** 7
**Confidence:** 2

**Review:**

The obtained result is valuable and can be accepted if the following problems can be clarified.
1. The paper should include comparisons against the existing literature to demonstrate its advantages.
2. The paper should be added to the Assumptions and definitions with relative references to show the rationality of this paper.
3. What examples or formulas are used in this paper to verify the effectiveness of the proposed method?

---

### Decision · Program_Chairs · 2024-09-06

Accept (Oral)